# The W.A.I.O.T. Definition of Peri-Prosthetic Joint Infection: A Multi-center, Retrospective Validation Study

**DOI:** 10.3390/jcm9061965

**Published:** 2020-06-23

**Authors:** Svetlana Bozhkova, Virginia Suardi, Hemant K Sharma, Hiroyuki Tsuchiya, Hernán del Sel, Mahmoud A. Hafez, Thami Benzakour, Lorenzo Drago, Carlo Luca Romanò

**Affiliations:** 1R.R. Vreden Russian Research Institute of Traumatology and Orthopaedics, S. Petersburg 195427, Russia; clinpharm-rniito@yandex.ru; 2RNIITO Department of Prevention and Treatment of Wound Infection, S. Petersburg 195427, Russia; 3Orthopedics Specialty School, University of Milan, 20100 Milano, Italy; virginia.suardi@unimi.it; 4Hull University Teaching Hospitals, Anlaby Road, Hull HU3 2JZ, UK; h.sharma@hull.ac.uk; 5Department of Orthopaedic Surgery, Graduate School of Medical Sciences, Kanazawa University, Kanazawa, Ishikawa 920-8641, Japan; tsuchi@med.kanazawa-u.ac.jp; 6Department of Orthopaedics, British Hospital Buenos Aires, Buenos Aires C1280, Argentina; hdelsel@argentina.com; 7Department of Orthopaedics, October 6 University, 12566 Cairo, Egypt; mhafez@msn.com; 8Zerktouni Orthopaedic Clinic, 20000 Casablanca, Morocco; t.benzakour@gmail.com; 9Clinical Microbiology, University of Milan, 20100 Milano, Italy; lorenzo.drago@unimi.it; 10Studio Medico Cecca-Romanò, Corso Venezia, 20121 Milano, Italy; 11Romano Institute, Rruga Ibrahim Rugova 1, 00100 Tirane, Albania

**Keywords:** PJI, definition, WAIOT, prosthesis, infection, diagnosis, hip, knee, validation

## Abstract

Peri-prosthetic joint infection (PJI) definition plays an important role in diagnostic and therapeutic decisions. However, while several criteria have been proposed by eminent institutions to define a PJI in the last decade, their clinical validation has been rarely performed. Aim of the present multicenter, international, retrospective study was to validate the World Association against Infection in Orthopedics and Trauma (WAIOT) pre/intra-operative PJI definition with post-operative confirmatory tests. A total of 210 patients, undergoing hip (*n* = 86) or knee (*n* = 124) revision surgery for any reason in six orthopedic centers in Africa, Asia, Europe and South America, were retrospectively evaluated at a two years minimum follow-up after surgery. All the available pre-, intra- and post-operative findings were collected and analyzed according to the WAIOT criteria, which include a set of tests to confirm (Rule In) or to exclude (Rule Out) a PJI. On average, patients were investigated pre/intra-operatively with 3.1 ± 1.1 rule out and 2.7 ± 0.9 rule in tests; the presence of a fistula or exposed implant was reported in 37 patients (17.6%). According to pre/intraoperative findings, 36.2% of the patients were defined as affected by high-grade PJI (*n* = 76; average score: 2.3 ± 0.8), 21.9% by low-grade PJI (*n* = 46; average score: 0.8 ± 0.8), 10.5% by biofilm-related implant malfunction (*n* = 22; average score: –1.6 ± 0.8), 2.9% as contamination (*n* = 6; average score: –3.5 ± 1.0), and 28.6% as no infection (*n* = 60; average score: –3.0 ± 1.4). Pre/intra-operative PJI definitions matched post-operative confirmatory tests, in 97.1% of the patients. This is, to our knowledge, one of the largest study ever conducted to validate a PJI definition The retrospective analysis in different centers was greatly facilitated by the structure of the WAIOT definition, that allows to include different tests on the basis of their sensitivity/specificity, while the comparison between pre/intra-operative and post-operative findings offered the internal validation of the scoring system. Our results authenticate the WAIOT definition as a reliable, simple tool to identify patients affected by PJI prior to joint revision surgery.

## 1. Introduction

Shared definitions of clinical conditions play a key role to compare the results of different trials and to eventually pool the data to perform systematic reviews and meta-analysis [1]. On the other hand, the definition of a clinical condition strongly contributes to the diagnostic and treatment decisions in any given patient.

In orthopedics, the selection of the relevant markers and tests needed to define a peri-prosthetic joint infection (PJI) has an impact on the choice of diagnostic procedures. These tests can be invasive, expensive and prone for further complications. Moreover, classifying an implant failure as a post-surgical infection has social, economic, psychological and medico-legal consequences [2].

Identifying PJI is crucial, yet there are no gold standard criteria available which will help in management of this complex and expensive revision surgery. In fact, the diagnosis of PJI remains challenging, with no single test providing absolute accuracy [3,4], and hence requiring a combination of clinical examination, serum and synovial fluid markers, imaging, microbiological and histological tests. Moreover, peri-prosthetic joint infections have a wide range of clinical presentations, from the acute, high-grade to the subclinical low-grade ones, [5,6] and even the most recent and complex scores may result as “inconclusive” [7]. Moreover, novel markers and diagnostic procedures are continuously discovered [8,9,10], while “older” investigations, like nuclear imaging, reported as sufficiently accurate by some specialists [11], are excluded by most of the current PJI definitions [12]. Due to lack of a 100% accurate test, at least six different definitions of PJI have been proposed in the last decade, providing a heterogeneous set of criteria, scoring systems and reference values [7,12,13,14,15,16,17].

In this complex panorama, even the validation of the existing definitions can be challenging. To the best of our knowledge, only one study has attempted to validate “The 2018 Definition of Peri-prosthetic Hip and Knee Infection” [7]. Soon thereafter, however, the definition validated in that study was further modified at the 2018 International Consensus Meeting (ICM) in Philadelphia. The modified version released by the ICM is substantially different from the clinically validated one (cf. Table 1).

In 2019, the World Association against Infection in Orthopedics and Trauma (WAIOT) proposed a novel approach to PJI definition [17] based on the relative ability of currently available tests to confirm (Rule In) or to exclude (Rule Out) a PJI (Table 2). To be included in the WAIOT definition, rule in and rule out tests should have, respectively, demonstrated specificity or sensitivity >90%, according to the available literature (cf. Table 3) [17]. Other features of the WAIOT definition are the inclusion of the information concerning the clinical presentation and history of the patient. WAIOT definition also proposes five distinct categories: high-grade and low-grade-PJI, biofilm-related implant malfunction, contamination and no infection. Furthermore, the WAIOT definition allows to confirm the pre/intra-operative definition with post-operative, findings, thus paving the way for a possible internal validation of pre/intra-operative definition to post-operative confirmation.

The aim of this multi-institutional, retrospective analysis was (1) to verify the clinical applicability of the WAIOT definition in a retrospective cohort of patients, admitted at orthopedic centers in different continents; (2) internal validation of the pre/intra-operative WAIOT definition with the post-operative confirmatory tests (3) confirm the patient classification and drive the surgical choice for revision surgery.

## 2. Material and Methods

After the institutional review board approval, we conducted a retrospective review of the medical records of 210 consecutive patients undergoing one- or two-stage revision total hip arthroplasty (THA) and total knee arthroplasty (TKA) for any reason in 6 orthopaedic centres (Buenos Aires, Argentina; Cairo, Egypt; Kanazawa, Japan; Milano, Italy; Saint Petersburg, Russia; Hull, UK) between January 2016 and December 2017. Each participating centre was required to provide the data of 30 to 40 patients.

Patients were classified pre/intra-operatively, according to the WAIOT definition, following evaluation of laboratory findings, imaging, histological and microbiological results. Laboratory and histopathology values were dichotomized as elevated or not based on the WAIOT cut-offs (cf. Table 3). Pre/intra-operative definition was validated by post-surgical histological and microbiological findings, as required by the WAIOT criteria [17].

According to the WAIOT definition, a minimum of two “rule out” and two “rule in” tests needs to be performed in any given patient, in order to define the presence of a peri-prosthetic joint infection.

WAIOT definition does not prescribe the tests and clinicians based on clinical, logistical and economic considerations, are free to decide the tests of their choice. In line with the definition requirements, patients without at least two pre/intra-operative rule in and two rule out tests and those without post-operative microbiological and histological confirmatory findings were excluded. In addition, patient undergoing partial prosthesis revision or revision surgery within three months from previous intervention were also excluded from the analysis.

### 2.1. Pre/Intra-Operative Assessment and Post-Operative Validation

A pre/intra-operative definition, incorporating all the relevant clinical findings and diagnostic tests as per Table 2 and Table 3, performed prior or during revision surgery, was formulated on the basis of the WAIOT criteria. In particular, the presence of a fistula or an exposed prosthesis and any positive rule in test is scored +1, while negative rule out tests are scored −1. No imputations are used for negative rule in tests, for positive rule out tests or for missing markers/tests.

Post-operative, confirmative definition was established by adding to the pre/intra-operative findings the histological and microbiological results on samples taken at surgery. In particular, cut-off for histology was according the modified Mirra’s criterion of at least 5 neutrophils in at least 3 high-power field section [17,18]. Three to 6 peri-prosthetic tissue samples were taken for microbiological testing; explanted biomaterials, including the prosthesis, modular parts and bone cement were sent for microbiological examination, using anti-biofilm mechanical or chemical processing techniques in 93/210 cases (44%). Cultures were maintained for 10 to 14 days. Even one positive sample was considered confirmatory for infection, in patients pre-operatively classified as high- or low-grade PJI or BIM. However, in patients pre-operatively defined as no infection, one positive sample was considered as contaminant [cf. 17].

The WAIOT definition allows to classify patients with a suspect PJI in five groups:

“high-grade PJI” (HG-PJI), in which the balance between positive “rule in” and negative “rule out” tests is ≥1 and two or more signs or symptoms of local inflammation (pain, swelling, redness, warmth, function laesa) are found. The HG-PJI is considered confirmed if post-operatively histological and/or at least one microbiological sample are positive.

“Low-grade PJI” (LG-PJI), in which the balance between positive “rule in” and negative “rule out” tests is ≥0 and, in a patient complaining of at least one of the followings: otherwise “unexplained” pain, swelling and/or reduced range of motion or functional impairment. The LG-PJI is confirmed post-operatively in presence of positive histology and/or at least one positive microbiological sample.

“Biofilm-related implant malfunction” (BIM), a condition similar to LG-PJI, but where the balance between positive “rule in” tests and negative “rule out” tests is 0. The diagnosis is made post-operatively, if a positive histology and/or if at least one microbiological sample is positive.

”No infection” is pre/intra-operatively defined, where surgical indication is other than infection, (e.g., wear debris, recurrent dislocation or joint instability, fracture, malposition, neuropathic pain, etc.). The balance between positive “rule in” and negative “rule out” tests is <0 and post-operative cultural and histological findings are all negative.

“Contamination” is said to occur in a condition similar to no infection, when a microorganism is recovered from a single synovial fluid or intra-operative sample, but histology is negative.

### 2.2. Statistical Analysis

Means and ranges were calculated with the Microsoft Excel 2013 (Microsoft, Redmond, WA, USA). Differences between groups were evaluated using a chi-squared test and Fisher’s exact test. The statistical significance was set at *p* < 0.05.

## 3. Results

Data for 210 patients (97 males and 113 females) from 6 orthopaedic centres was analysed. Average age at the time of surgery was 63.9 ± 10.9 years. At revision, time from the most recent surgery was 2.1 ± 1.9 years (range 0.5 to 6 years).

Seventy-six (36.2%) of the patients were classified as high-grade PJI (HG-PJI), 46 (21.9%) as low-grade PJI (LG-PJI), 22 (10.5%) as biofilm-related implant malfunction (BIM), 60 (29%) as no infection and 6 (2.8%) as contamination.

A total of 124 (59.1%) patients had a knee prosthesis and 86 (40.9%) a hip joint arthroplasty. There was no statistical difference between patients with hip or knee implants classified as infected or not, according to the WAIOT definition (*p* = 0.3). 

Ninety-one (43.3%) patients had a one-stage revision, while the remaining 119 (56.7) two-stage protocol. Two-stage procedures were most commonly performed in patients that were defined as having an infection using the WAIOT criteria (102/144, 70.8%), compared to patients defined as not infected (17/66, 25.7%) (*p* < 0.0001) (Table 4).

Excluding the presence of a fistula, a total of 560 rule in and 717 rule out tests were performed, with an average 2.7 ± 0.9 and 3.1 ± 1.1 tests per patient, respectively.

Pre/intra-operative score of patients classified as HG-PJI ranged from 1 to 4 with an average score of 2.3 ± 0.8; Low-Grade PJIs displayed an average score of 0.8 ± 0.8 which reflects a nearly perfect balance between positive and negative tests in these patients. Biofilm-related implant malfunction patients had an average score of –1.6 ± 0.8, as the majority of the tests were negative, and thereby emphasizing the difficulties surgeon may face in diagnosis. As expected, patients classified as No Infection or as Contamination, had a majority of tests negative, with an average score of, respectively, –3.0 ± 1.4 and –3.5 ± 1.0 (Table 4).

Table 5 reports the absolute number of tests found positive in each class of patients and their relative percentage. Physical examination was performed in all cases: the presence of a fistula was recorded in 37 out of 210 patients (17.6%) and, more specifically in 39.5% of the patients that were defined as high-grade PJI, in 10.9% of those defined as low-grade PJI and in two patients (9.1%) classified as biofilm-related implant malfunction. Most often reported tests were C-reactive protein and ESR, performed in 210 (100%) and 186 (88.6%) patients, respectively. Other serum markers, D-dimer (69 cases; 32.9%), procalcitonin (10; 4.8%) and interleukin-6 (IL-6, 1; 0.5%) were much less frequently reported. Most common synovial fluid tests carried out were cultural examination, leukocyte esterase and white blood cell count, 127 (60.5%), 133 (63.3%) and 107 (51%) patients, respectively. Alpha-defensin was reported in 42 (20%) cases. Tc99 bone scans were available in 39 (18.6%) and combined leukocyte bone marrow scintigraphy in 38 (18.1%) of cases. Frozen sections were reported in 33 (15.7%) of cases. Overall, performed tests were found positive in 80.0% of HG-PJI, 57.5% of LG-PJI, 27.0% of BIM, 17.4% of contamination and 10.0% of no infection patients. Excluding the presence of a sinus, rule in tests were positive in 83.9% of HG-PJI, 50.4% of LG-PJI, 16.0% of BIM, 11.1% of contamination and 8.1% of no infection cases. Conversely, rule out tests were negative in 9.5%, 23.7%, 59.4%, 77.8% and 86.0% of the patients categorized as HG-PJI, LG-PJI, BIM, contamination and no infection, respectively.

Overall, post-operative findings confirmed pre/intra-operative definitions in 204/210 cases (97.1%). One no infection and 5 LG-PJI definitions were not confirmed post-operatively (Table 6). The patient considered as no infection had been treated with a one-stage knee revision procedure; his pre-operative serum CRP and ESR were both negative, as well as the synovial white blood cell count (1.350 cells/uL) and the leukocyte esterase. Post-operative findings gave conflicting results, with all intra-operative cultural examinations negative, but with positive histology (Table 7). In all 5 unconfirmed LG-PJI cases post-operative histological and microbiological findings were negative. Three patients were on antibiotics therapy at the time of surgery. These 5 patients had mixed positive and negative pre-operative findings, with a pre/intra-operative score ranging from 0 to 2. Three patients also had positive cultures on pre-operative synovial fluid aspiration (*Staph. epidermidis* in two cases and *Staph. capitis* in the third patient) (Table 7).

Table 8 reports the results of the post-operative cultural examinations. Overall, 144 microorganisms were isolated post-operatively. A mixed flora was found in 14 patients, while in 84 cases cultural examinations gave negative results (60/60 in patients defined as no infection, 5/22 in those classified as BIM, 17/47 and 2/76 in LG-PJI and HG-PJI, respectively). Patients affected by LG-PJI and BIM had significantly more coagulase-negative *Staphylococci* than *Staphylococcus aureus* isolates, compared to HG-PJI (*p* = 0.01); moreover, LG-PJIs and BIMs displayed significantly more negative cultural examinations than those observed in patients defined as HG-PJI (*p* < 0.0001 and *p* = 0.005, respectively).

## 4. Discussion

This multi-institutional and international study shows that pre/intra-operative definition of patients with suspect PJI can reliably be confirmed using the WAIOT criteria. This is, to our knowledge, the largest retrospective study performed to clinically validate the WAIOT PJI definition and, more generally, one of the largest for validation of any PJI definitions.

Diagnostic definitions of PJI has continuously evolved in the last decade [19]. However, with the exception of the latest Musculoskeletal Infection Society (MSIS) criteria [7] none of the current PJI definitions have been validated in clinical trials. In this large retrospective and partly prospective multi-institutional study, the authors aimed at updating the previous MSIS PJI definition, released a few years before, [15] introducing novel markers and a new scoring system. However, the clinically validated score was modified few months after the conclusion of the clinical trial, at the Consensus Meeting in Philadelphia in 2018, [12] making the validation of the score ineffectual. Consensus ICM 2018 meeting, in the modified definition, scored synovial fluid markers differently and the overall definition of infection according to the “minor criteria” was changed (cf. Table 1). Furthermore, the agreement on the definition between the experts at the Consensus was relatively low 68%. [12].

Our findings point out the unique ability of the WAIOT definition to reliably differentiate five categories, namely high- and low-grade infections, biofilm-related implant malfunction, contamination and no infection. This distinction reflects and translates for the first time into the clinical practice of the new category and understanding of the biofilm-related nature of implant-related infections [6,20]. This highlights the capability of bacteria to persist embedded in biofilms [21] or even inside the host’s cells, [22] without inducing significant inflammatory or immunological reactions and explains the difficulties to detect peri-prosthetic infections with the usual diagnostic tests and cut-off values [23,24]. WAIOT classification identifies different possible clinical scenarios, which will not only improve our diagnostic suspicion and understanding the limits of current tests, but also to design studies targeted at treatments for specific patients’ populations. Low-grade infections and biofilm-related implant malfunction may also prompt clinicians to meticulously investigate the relatively frequent “unexplained” painful joint prosthesis, affecting 5 to 10% of patients undergoing joint replacement surgeries [25]. Diagnostic challenges in PJI are emphasized in low-grade infections with a nearly perfect balance between positive and negative tests and biofilm-related implant malfunction with the prevalence of tests excluding pre-operatively infection [5,26].

In particular, our findings show that approximately half of the patients classified as infected should be considered difficult-to-diagnose; in fact, 46 (31.9%) and 22 (15.3%) patients were classified, respectively, as low-grade infection or biofilm-related implant malfunction. High-grade infection accounted only for the remaining 76 (52.7%) patients, while a sinus tract could only be observed in 37 (25.7%) subjects. This observation further points out the need for a PJI definition that takes into account the frequent occurrence, in the clinical practice, of implant-related infections without the classical inflammatory signs.

The present study challenged pre/intra-operative tests with post-operative confirmatory findings. This was possible due to the specific structure of the WAIOT definition, which distinguishes a pre/intra-operative definition from the post-operative histological and microbiological confirmatory findings. The present study evaluated and was able to differentiate “infection” from “no infection”. This study also validated that patients with conflicting or even negative markers and pre/intra-operatively classified as low-grade PJIs or BIMs, were infected according to post-operative results.

As expected, low-grade infections were arduous to diagnose, with five cases showing all negative post-operative findings. This observation strongly supports the need to improve both histological and microbiological procedures, especially in patients with expected slow-growing microorganisms [27,28,29,30]. The analysis of cultural examinations points out how slow-growing microorganisms are more often found in conditions classified as low-grade Infections or biofilm-related implant malfunctions. Both of which are also associated with an increasing proportion of negative cultures, compared to high-grade infections. To refine microbiological analysis, the WAIOT definition recommends using anti-biofilm processing techniques, prolonged cultures and stopping any antibiotic treatment prior to surgery [17,30,31,32]. Given the retrospective nature of the present analysis, these recommendations were not fulfilled and only approximately half of the patients were investigated with anti-biofilm techniques. Genomic pathogen identification is another promising, yet not readily available, tool to improve pathogen identification [9,33].

The present study also allowed to test for the first time the concept that different diagnostic tests can be alternatively used, provided that they have a similar sensitivity or specificity. As previously reported, [17] the WAIOT definition, at variance with other classification systems, selects the rule in and rule out tests on the basis of their specificity and sensitivity, which must exceed 90%, according to the available literature. This uniqueness of the WAIOT definition also greatly facilitated this multi-center, retrospective study, that would have been otherwise much more difficult to perform with other definitions, that impose a specific set of tests to be performed in each patient. The WAIOT definition relies on simple routine tests which can be performed in any hospital, depending on resources available and local policies, and thereby is very versatile diagnostic criteria.

The present analysis also contributes to verify the hypothesis that a minimum of two rule in and 2 rule out tests (excluding the presence of the fistula) may be sufficient to identify a PJI. Our analysis shows that the average of 2.7 rule in and 3.1 rule out tests per patient provided enough pre/intra-operative findings to define a PJI in the studied population (cf. Table 4). 

Although not evidenced by the present study, it should be noted, that several biomarkers can be falsely positive for PJI, in patients with concurrent local or systemic inflammatory conditions (e.g., rheumatological disease, pneumonia, acute urinary tract infections, deep vein thrombosis, etc.). [34] Hence, a particular attention should be payed to patients with positive rule in tests, when “One or more condition(s), other than infection, can cause the symptoms or the reason for reoperation” (cf. Table 2) In these patients, more than 2 rule in and 2 rule out tests should be performed and the tests should be repeated, to evaluate their pattern over time or once the confounding inflammatory condition has settled.

Presence of a sinus tract, according to the WAIOT definition is considered as a rule in finding, which, if present should be scored as +1. In our series, a fistula or an exposed implant was observed in 37/210 patients or in 37/144 (25.7%) of infected cases (cf. Table 5); all 37 patients were defined as infected, according to the WAIOT definition. More specifically, 30/37 were classified as High-Grade PJI, 5/37 as Low-Grade and 2/37 as Biofilm Related Implant Malfunction (39.5%, 10.9% and 9.1% of all HG-PJI, LG-PJI and BIM, respectively). Our findings support that patients presenting with a draining sinus or an exposed implant can be reliably defined as PJI only on the basis of this clinical sign. Although in isolated reports a sinus tract has been associated with conditions other than infection, [35] the presence of a fistula or of an exposed implant is generally considered highly specific for PJI. [36] Moreover, some inflammatory markers may be falsely negative in patients with infection and a draining fistula. [37,38,39,40,41] Considering all of the above, a modification of the original WAIOT definition can be envisaged, that includes the presence of a sinus tract or of an exposed implant as a pathognomonic sign of PJI (cf. Table 9 and Table 10). While this change does not affect the results of the present analysis, it leads to a significant simplification of the definition and prevents possible confusion due to the interpretation of rule in and rule out tests in the event of such clinical findings.

## 5. Study Limitations

The present clinical investigation has a number of limitations. First of all, the retrospective and multi-institutional nature of the study may have introduced some bias, including missing data and different procedures (e.g., microbiological and histological examinations). This limit may be further appreciated when considering semi-quantitative tests (e.g., leukocyte esterase strips) or other examinations, which are subject to interpretation, like nuclear imaging. Although combined leukocyte bone scan is generally considered highly specific to diagnose PJI [42], and hence a reliable rule in test, in the present analysis it did provide an unexpected high rate of false positive results, even if in a very limited series of patients (cf. Table 5). These findings may not be regarded as a demonstration of the lack of accuracy of leukocyte bone scan, since our study was neither designed nor powered to assess this; however, these results point out the possible impact of a lack of standard criteria across centers to perform and interpret bone scans.

Present analysis being retrospective may suffer from “selection bias”. This may explain the relatively high number of patients classified as infected in the studied population. Several patients, undergoing revision surgery for reasons other than infection, like prosthetic components wear, recurrent dislocation, peri-prosthetic fracture, etc. may have been excluded by the present analysis, because they did not meet the minimum required pre- and intra-operative investigations (two rule in and two rule out tests and cultural and histological examinations at surgery). This limitation is shared by other studies devoted to assess PJI definition validity [7] and may only be overcome by a prospective study on consecutive patients.

Results of the present analysis may not be valid for patients with an interval of less than 90 days from index surgery, as this was another exclusion criterium. A recent surgical procedure may affect the cut-offs and the overall accuracy of serum and synovial biomarkers, as well as nuclear imaging.

Another limitation concerns the relatively low number of patients included in the analysis and the fact that each center contributed with a limited sample of 30 to 40 cases, which may not be true representation of the patients’ population undergoing revision surgery. Although this is one of the largest studies ever performed in the field, the data should be confirmed in larger cohorts of patients and, possibly, by prospective trials.

In conclusion, its limits notwithstanding, this retrospective, multicenter and international study shows that the WAIOT definition of PJI allows to reliably discriminate infection from no infection in patients undergoing joint revision surgery, at a minimum of 90 days from the previous surgery. The present analysis illustrates the ability of the pre/intra-operative classification to match post-operative confirmatory results. This validation is particularly relevant in patients with conflicting or even negative markers and patients which pre/intra-operatively are classified as low-grade PJIs or BIMs. Moreover, based on our results, a simplification of the definition system can be proposed, according to which the presence of a sinus or an exposed implant is regarded as a pathognomonic sign of infection.

Further prospective studies are warranted to confirm and expand the results of this study.

## Figures and Tables

**Table 1 jcm-09-01965-t001:** Comparison of the “Musculoskeletal Infection Society (MSIS) updated criteria for PJI”, validated in a clinical trial, and the “Proposed 2018 ICM Criteria for PJI” finally voted and released at the International Consensus Meeting (ICM) of Philadelphia in 2018.

MSIS 2018 Criteria for PJI [7]	ICM Criteria for PJI [12]
**Major Criteria** (at least one of the following)	**Major Criteria** (at least one of the following)
Two positive growths of the same organism; Sinus tract with evidence of communication to the joint or visualization of the prosthesis	Two positive growths of the same organism using standard culture methods; Sinus tract with evidence of communication to the joint or visualization of the prosthesis
Decision: Infected	Decision: Infected
**Minor Criteria**	**Minor Criteria**
Preoperative Diagnosis: (a) Elevated serum CRP or D-Dimer (score 2) (b) Elevated serum ESR (score 1) (c) Elevated synovial WBC count or LE (score 3) (d) Positive alpha-defensin (score 3) (e) Elevated synovial PMN% (score 2) (f) Elevated synovial CRP (score 1)	Combined preoperative and postoperative score: (a) Elevated serum CRP or D-Dimer (score 2) (b) Elevated serum ESR (score 1) (c) Elevated synovial WBC count or LE or Positive alpha-defensin (score 3) (d) Elevated synovial PMN% (score 2) (e) Single positive culture (score 2) (f) Positive histology (score 3) (g) Positive intraoperative purulence (score 3)
Decision:	Decision:
≥6 Infected	≥6 Infected
2–5 Possibly Infected (“Consider further molecular diagnostics such as next-generation sequencing”)	3–5 Inconclusive (“Consider further molecular diagnostics such as next-generation sequencing”)
0–1 Not Infected.	<3 Not Infected.
Inconclusive pre-op score or dry tap (a) Preoperative score (score -) (b) Positive histology (score 3) (c) Positive purulence (score 3) (d) Single positive culture (score 2)	
Decision: ≥6 Infected 4–5 Inconclusive (“Consider further molecular diagnostics such as next-generation sequencing”) <3 Not Infected.	

Abbreviations: MSIS: Musculoskeletal Infection SocietyICM: International Consensus Meeting; ESR: serum erythrocyte sedimentation rate; CRP: Serum C-reactive protein; PNM%: Synovial neutrophil percentage; WBC: White blood cells; LE: Leukocyte esterase.

**Table 2 jcm-09-01965-t002:** World Association against Infection in Orthopedics and Trauma (WAIOT) Definition of Prosthetic Joint Infection [17].

	No Infection	Contamination	BIM	LG-PJI	HG-PJI
Clinical presentation	One or more condition(s), other than infection, can cause the symptoms or the reason for reoperation (e.g., wear debris, metallosis, recurrent dislocation or joint instability, fracture, malposition, neuropathic pain)		One or more of the followings: Otherwise “unexplained” pain, swelling, stiffness		Two or more of the followings: Pain, swelling, redness, warmth, *functio laesa*
Number of Positive Rule In Minus Number of Negative Rule Out tests	<0	<0	<0	≥0	≥1
Post-Operatively Confirmed If	Negative cultural examination	One pre- or intra-operative positive culture, with negative histology	Positive cultural examination (preferably with anti-biofilm techniques) and/or positive histology		

Abbreviations: WAIOT: World Association against Infection in Orthopedics and Trauma; BIM: Biofilm-related implant malfunction; lg-pji: Low-grade peri-prosthetic joint infection; hg-pji: High-grade peri-prosthetic joint infection.

**Table 3 jcm-09-01965-t003:** Pre/intra-operative tests used to exclude (“rule out”) or to confirm (“rule in”) a PJI according to WAIOT definition, with the respective cut-off values [17].

**Rule OUT Tests** **Each Negative Test Scores −1 (Positive Rule Out Tests Score 0)**	
Serum	ESR (>30 mm/h) CRP (>10 mg/L)
Synovial fluid	WBC (>1500/μL) LE (++) Alpha-Defensin immunoassay (>5.2 mg/L)
Imaging	Tc99 bone scan
**Rule IN Tests** **Each Positive Test Scores +1 (Negative Rule In Tests Score 0)**	
Clinical examination	Draining sinus or exposed joint prosthesis
Serum	IL-6 (>10 pg/mL) PC (>0.5 ng/mL) D-Dimer (>850 ng/mL)
Synovial fluid	Cultural examination WBC (>3000/mL) LE (++) Alpha-defensin immunoassay (>5.2 mg/L) or lateral flow test
Imaging	Combined leukocyte and bone marrow scintigraphy
Histology	Frozen section (5 neutrophils in at least 3 HPFs)

Abbreviations: ESR: Erythrocyte sedimentation rate; CRP: C-reactive protein; IL-6: Interleukin-6; WBC: White blood cell count; PC: Procalcitonin; LE: Leukocyte esterase strip (++); HPFs: High power fields (×400).

**Table 4 jcm-09-01965-t004:** Pre/Intra-operative data of the studied population.

	Patients	Hip Prosthesis	Knee Prosthesis	One-Stage Revision	Two-Stage Revision	Number of Performed Rule in Tests	Number of Performed Rule in Tests Per Patient	Number of Performed Rule out Tests	Number of Performed Rule out Tests Per Patient	Pre/Intra-Operative Score
	(*n* and %)	(*n* and %)	(*n* and %)	(*n* and %)	(*n* and %)	(*n*)	(mean ± SD and range)	(*n*)	(mean ± SD and range)	(mean ± SD and range)
HG-PJI	76 (36.2)	37 (48.7)	39 (51.3)	19 (25.0)	57 (75)	193	2.5 ± 0.9 (2–6)	222	2.5 ± 1.0 (2–5)	2.3 ± 0.8 (1–4)
LG-PJI	46 (21.9)	19 (41.3)	27 (58.7)	15 (32.6)	31 (67.4)	113	2.5 ± 0.7 (2–4)	156	3.0 ± 1.2 (2–5)	0.8 ± 0.8 (0–2)
BIM	22 (10.5)	6 (27.3)	16 (72.7)	8 (36.4)	14 (63.6)	50	2.3 ± 0.7 (2–4)	69	3.1 ± 1.2 (2–5)	−1.6 ± 0.8 (–3–−1)
Contamination	6 (2.9)	1 (16.7)	5 (83.3)	6 (100)	0 (0)	18	2.8 ± 1.0 (2–4)	27	4.3 ± 0.8 (3–5)	–3.5 ± 1.0 (–5–−2)
No Infection	60 (28.6)	23 (38.3)	37 (61.7)	43 (71.7)	17 (28.3)	186	3.1 ± 0.9 (2–5)	243	3.9 ± 1.2 (2–5)	–3.0 ± 1.4 (–5–−1)
TOTAL	210 (100)	86 (40.9)	124 (59.0)	91 (43.3)	119 (56.7)	560	2.7 ± 0.9 (2–6)	717	3.1 ± 1.1 (2–5)	–0.1 ± 2.5 (–5–4)

Abbreviations: HG-PJI: High-grade peri-prosthetic joint infection; LG-PJI: Low-grade peri-prosthetic joint infection; BIM: Biofilm-related implant malfunction.

**Table 5 jcm-09-01965-t005:** Tests found positive in the five groups of patients, classified according to the WAIOT definition.

		HG-PJI (*n* = 76)		LG-PJI (*n* = 46)		BIM (*n* = 22)		Contamination (*n* = 6)		No infection (*n* = 60)	
		Positive Tests/Performed Tests	%	Positive Tests/Performed Tests	%	Positive Tests/Performed Tests	%	Positive Tests/Performed Tests	%	Positive Tests/Performed Tests	%
Physical examination	Fistula	30/76	39.5	5/46	10.9	2/22	9.1	0/6	0	0/60	0
Serum markers	CRP > 10 mg/L	71/76	93.4	37/46	80.4	5/22	22.7	1/6	16.7	10/60	16.7
	ESR > 30 mm/hr	56/62	90.3	37/45	82.2	10/19	52.6	1/6	16.7	11/54	20.4
	IL-6 > 10 pg/mL	1/1	100	0/0	n/a	0/0	n/a	0/0	n/a	0/0	n/a
	PC > 0.5 ng/mL	4/4	100	0/1	0.0	0/1	0.0	0/0	n/a	0/4	0.0
	D-dimer > 850 ng/mL	21/31	67.7	2/14	14.3	0/9	0.0	0/2	0.0	0/13	0.0
Synovial fluid markers	WBC > 3000/mcL	22/25	88.0	16/23	69.6	0/7	0.0	0/5	0.0	2/47	4.3
	WBC > 1500/mcL	24/25	96.0	17/23	73.9	5/7	71.4	1/5	20.0	2/47	4.3
	LE ++	36/43	83.7	11/23	47.8	2/13	15.4	0/6	0.0	3/48	6.3
	Alpha-defensin >5.2 mg/L or lateral flow test	8/9	88.9	1/4	25.0	0/2	0.0	0/2	0.0	0/25	0.0
Imaging	Tc99 Bone scan	6/7	85.7	15/15	100	6/6	100	2/2	100	9/9	100
	Combined Leukocyte/ Bone Marrow scintigraphy	15/16	93.8	12/14	85.7	1/3	33.3	1/1	100	3/4	75.0
Microbiology	Cultural examination	34/42	81.0	15/31	48.4	4/14	28.6	0/2	0.0	7/38	18.4
Histology	Frozen sections	21/22	95.5	0/3	0.0	1/1	100	0/0	n/a	0/7	0.0
TOTAL excl. fistula		363/415	87.5	176/269	65.4	36/119	30.3	9/45	20.0	49/429	11.4
TOTAL		303/491	80.0	181/315	57.5	38/141	27.0	9/51	17.6	49/489	10.0

Abbreviations: HG-PJI: High-grade peri-prosthetic joint infection; LG-PJI: Low-grade peri-prosthetic joint infection; BIM: Biofilm-related implant malfunction; ESR: Erythrocyte sedimentation rate; CRP: C-reactive protein; IL-6: Interleukin-6; WBC: White blood cell count; PC: Procalcitonin; LE: Leukocyte esterase strip.

**Table 6 jcm-09-01965-t006:** Pre/Intra-operative patients’ classification and post-operative confirmatory results.

	Patients	Pre/Intra-Operative Score	Post-Operative Positive Cultural Examination	Post-Operative Positive Histology	Post-Operative Confirmed Pre/Intra-Operative Definition
	(*n* and %)	(mean ± SD and range)	(Positive tests/tests performed and %)	(Positive tests /tests performed and %)	(*n* and %)
HG-PJI	76 (36.2)	2.3 ± 0.8 (1–4)	74/76 (97.3)	41/48 (85.4)	76/76 (100)
LG-PJI	46 (21.9)	0.8 ± 0.8 (0–2)	29/46 (63.0)	23/34 (67.6)	41/46 (89.1)
BIM	22 (10.5)	–1.6 ± 0.8 (–3–1)	17/22 (77.3)	9/16 (57.2)	22/22 (100)
Contamination	6 (2.9)	–3.5 ± 1.0 (–5–2)	6/6 (100)	0/3 (0)	6/6 (100)
No Infection	60 (28.6)	–3.0 ± 1.4 (–5–1)	0/60 (0)	1/46 (2.2)	59/60 (98.3)
TOTAL	210(100)	–0.1 ± 2.5 (–5–4)	126/210 (60.0)	74/147 (50.3)	204/210 (97.1)

Abbreviations: HG-PJI: High grade peri-prosthetic joint infection; LG-PJI: Low-grade peri-prosthetic joint infection; BIM: Biofilm-related implant malfunction.

**Table 7 jcm-09-01965-t007:** Tests results of post-operatively not confirmed pre/intra-operative definitions. One patient (Case 1) defined as No Infection did show a positive histological finding, while the remaining 5 patients (Cases 2 to 6) were classified as low-grade PJI based on pre-operative tests, but were all negative at histological and cultural examination at surgery.

	Patient		1	2	3	4	5	6
Clinical examination	Fistula		Negative	Negative	Negative	Negative	Negative	Negative
Serum	CRP ESR		Negative	Positive	Positive	Negative	Positive	Negative
			Negative	Positive		Negative	Positive	Positive
Synovial fluid	WBC LE		Negative		Positive		Negative	Positive
			Negative		Negative	Positive	Positive	Positive
	Cultural examination			Positive		Positive	Positive	
	Alpha-defensin							Negative
Imaging	Tc99 Bone scan					Positive	Positive	
	Leukocyte/Bone Marrow scintigraphy			Positive		Positive		
Positive rule in tests Score			0	2	1	3	2	2
Negative rule out tests Score			–4	0	–1	–2	–1	–2
Pre/intra-operative Score			–4	2	0	1	1	0
Pre/intra-operative Definition			No Infection	LG-PJI	LG-PJI	LG-PJI	LG-PJI	LG-PJI
Surgical procedure			One-stage knee revision	Two-stage knee revision	Two-stage hip revision	Two-stage knee revision	Two-stage knee revision	
Post-operative cultural examination			Negative	Negative	Negative	Negative	Negative	Negative
Post-operative histology			Positive	Negative	Negative	Negative	Negative	Negative

Abbreviations: LG-PJI: Low-grade peri-prosthetic joint infection; ESR: Erythrocyte sedimentation rate; CRP: C-reactive protein; WBC: White blood cell count; LE: Leukocyte esterase strip.

**Table 8 jcm-09-01965-t008:** Post-operative results of cultural examination.

	HG-PJI	LG-PJI	BIM	Contamination	No Infection
	*n* = 76	*n* = 46	*n* = 22	*n* = 6	*n* = 60
	(*n* and %)	(*n* and %)	(*n* and %)	(*n* and %)	(*n* and %)
Coagulase-negative *Staphylococci*	28 (31.5)	15 (53.6)	11 (55)	3 (42.9)	
*Staphylococcus aureus*	25 (28.1)	4 (14.3)		1 (14.3)	
*Streptococci* spp.	12 (13.5)	1 (3.6)			
*Enterococci* spp.	6 (6.7)	1 (3.6)			
*Candida* spp.	4 (4.5)	1 (3.6)			
*Pseudomonas aeruginosa*	3 (3.4)	3 (10.7)			
*Propionibacterium acnes*	2 (2.2)	2 (7.1)	5 (25)	1 (14.3)	
*Escherichia coli* (1 ESBL +)	3 (3.3)		1 (5)		
*Citrobacter koseri*	2 (2.2)				
*Enterobacter*	1 (1.1)				
*Klebsiella oxytoca*	1 (1.1)				
*Bacillus lentus*			1 (5)		
*Corynebacterium*	1 (1.1)		1 (5)	1 (14.3)	
*Micrococci*	1 (1.1)	1 (3.6)	1 (5)	1 (14.3)	
TOTAL	89 (100)	28 (100)	20 (100)	7 (100)	
Patients with a mixed flora	10 (13.2)	0	3 (13.6)	1 (16.7)	
Patients with negative cultures	2 (2.6)	17 (37)	5 (22.7)	0	60 (100)

**Table 9 jcm-09-01965-t009:** Modified WAIOT Definition of peri-prosthetic joint infection (PJI). The presence of a sinus or an exposed implant is considered as a pathognomonic sign of infection.

	No Infection	Contamination	BIM	LG-PJI	HG-PJI
Clinical presentation	One or more condition(s), other than infection, can cause the symptoms or the reason for reoperation (e.g., wear debris, metallosis, recurrent dislocation or joint instability, fracture, malposition, neuropathic pain)		Sinus tract or exposed implant, or ≥1 of the followings: otherwise “unexplained” pain, swelling, stiffness		Sinus tract or exposed implant or ≥2 of the followings: pain, swelling, redness, warmth, *Functio laesa*
Positive rule in minus Negative rule out tests	<0	<0	<0	≥0	≥1
Post-operatively confirmed if	Negative cultural examination	One pre- or intra-operative positive culture, with negative histology	Positive cultural examination (preferably with anti-biofilm techniques) and/or positive histology		

Abbreviations: WAIOT: World Association against Infection in Orthopedics and Trauma; BIM: Biofilm-related implant malfunction; LG-PJI: Low-grade peri-prosthetic joint infection; HG-PJI: High-grade peri-prosthetic joint infection.

**Table 10 jcm-09-01965-t010:** Rule In and Rule Out tests of the modified WAIOT PJI.

Definition.	Rule In Tests	Rule Out Tests
Clinical examination	Draining sinus or exposed joint prosthesis *	
Serum	IL-6 (>10 pg/mL) ** PC (>0.5 ng/mL) ** D-Dimer (>850 ng/mL) **	ESR (>30 mm/h) *** CRP (>10 mg/L) ***
Synovial fluid	Cultural examination ** WBC (>3000/mL) ** LE (++) ** AD immuunoassay (>5.2 mg/L) or lateral flow test **	WBC (>1500/μL) *** LE (++) *** AD immunoassay (>5.2 mg/L) ***
Imaging	Combined leukocyte and bone marrow scintigraphy **	Tc99 bone scan ***
Histology	Frozen section (5 neutrophils in ≥3 HPFs **	

***** If positive, consider as infected; ** Positive Test Scores +1; *** Negative Test Scores −1. Abbreviations: WAIOT: World Association against Infection in Orthopedics and Trauma; ESR: Erythrocyte sedimentation rate; CRP: C-reactive protein; IL-6: Interleukin-6; WBC: White blood cell count; PC: Procalcitonin; LE: Leukocyte esterase strip (++); AD: Alpha-defensin; HPFs: High power fields (×400).

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
