# Peer review of "The W.A.I.O.T. Definition of Peri-Prosthetic Joint Infection: A Multi-center, Retrospective Validation Study"

_jcm, 2020, doi:10.3390/jcm9061965_

Round 1

Reviewer 1 Report

Congratulations on this interesting work. I have only some minor issues to raise:

  • Please have the text edited by a native speaker, some minor mistakes can be found that impair the understanding of the content
  • As the data was collected in part pro- and in part retrospectively, please try to make a clearer distinction in the Methods and the Results section. Maybe even separate the two data modalities and show results for each?
  • I have noticed a high proportion of high-grade infections. However, these are rather easy to diagnose in practice and do not pose significant problems even using the other definitions. I suggest a separate analysis of only the low-grade, BIM and not infected groups to show the diagnostic value of your method.
  • Same goes for the high rate of patients with a fistula. Both points should be discussed more intensely.
  • The poor performance of imaging is an interesting finding. Please discuss this further. I am not sure whether it can be pinned on just the inter-observer reliability (if so - imaging should not be included in the WAIOT criteria). Could the publications leading to inclusion in the WAIOT criteria have been biased? Or driven by particular interests in the imaging community?

Author Response

Congratulations on this interesting work. I have only some minor issues to raise:

THANK YOU FOR TAKING THE TIME TO REVIEW OUR PAPER.

  • Please have the text edited by a native speaker, some minor mistakes can be found that impair the understanding of the content

DONE, THANK YOU.

  • As the data was collected in part pro- and in part retrospectively, please try to make a clearer distinction in the Methods and the Results section. Maybe even separate the two data modalities and show results for each?

ALL THE DATA WERE COLLECTED ONLY RETROSPECTIVELY. THIS IS MADE CLEAR IN THE TITLE AND IN THE INTRODUCTION (LINE 89-90) AND METHODS SECTIONS (LINE 96).

FURTHERMORE, THE RETROSPECTIVE NATURE OF THE STUDY IS DISCUSSED AS A STUDY LIMITATION IN THE DISCUSSION SECTION (LINE 303)

  • I have noticed a high proportion of high-grade infections. However, these are rather easy to diagnose in practice and do not pose significant problems even using the other definitions. I suggest a separate analysis of only the low-grade, BIM and not infected groups to show the diagnostic value of your method.

OVERALL, 144 PATIENTS WERE CLASSIFIED AS INFECTED. ONLY APPROXIMATELY HALF OF THEM WERE HIGH-GRADE INFECTIONS (76, 52.7%), WHILE THE REMAINING WERE LOW-GRADE (46, 31.9%) OR BIM (22, 15.3%). AS THE REVIEWER CORRECTLY POINTS OUT THIS FACT IS OF MAJOR IMPORTANCE, SINCE THESE ARE THE PATIENTS THAT MOST OFTEN CAN BE MISSED BY CURRENT PJI DEFINITIONS.

THIS ISSUE HAS NOW BEEN UNDERLINED IN THE DISCUSSION SECTION (LINE 246-252).

  • Same goes for the high rate of patients with a fistula. Both points should be discussed more intensely.

YES, THIS IS NOW ALSO MORE EXTENSIVELY DISCUSSED AT (LINE 246-252).

  • The poor performance of imaging is an interesting finding. Please discuss this further. I am not sure whether it can be pinned on just the inter-observer reliability (if so - imaging should not be included in the WAIOT criteria). Could the publications leading to inclusion in the WAIOT criteria have been biased? Or driven by particular interests in the imaging community?

THIS IS ANOTHER VERY IMPORTANT POINT.

GIVEN THE RETROSPECTIVE NATURE OF THE PRESENT STUDY, THE LACK OF STANDARDIZED TECHNIQUES AND INTERPRETATION CRITERIA OF IMAGING TECHNIQUES IS A POSSIBLE SOURCE OF BIAS. THIS IS DISCUSSED AT LINE 311-319.

AS REPORTED IN THE DISCUSSION, HOWEVER, THIS STUDY WAS NOT INTENDED OR POWERED TO ASSESS THE RELATIVE ACCURACY OF NUCLEAR IMAGING TECHNIQUES (OR OF OTHER DIAGNOSTIC TESTS); HENCE, WE MAY NOT, ONLY ON THE BASIS OF OUR FINDINGS, CONTRADDICT PUBLISHED META-ANALYSIS THAT FORM THE BASIS OF THE WAIOT DEFINITION CRITERIA AND THAT DO REPORT QUITE A HIGH SENSITIVITY OF BONE SCAN AND HIGH SPECIFICITY OF LEUKOCYTE SCAN (CF. https://link.springer.com/article/10.1007/s11999-016-5218-0).

THE VALUE OF NUCLEAR IMAGING TO DIAGNOSE PJI HAS ALSO BEEN RECENTLY STATED BY A CONSENSUS PAPER, RELEASED BY THE EUROPEAN ASSOCIATION OF NUCLEAR MEDICINE (EANM), EUROPEAN BONE AND JOINT INFECTION SOCIETY (EBJIS) AND ECCMID, WHERE STANDARDS TO PERFORM AND INTERPRET THE RESULTS OF NUCLEAR IMAGING ARE ALSO REPORTED. (https://pubmed.ncbi.nlm.nih.gov/30683987/ ).

MORE RECENTLY, WE HAVE ADDRESSED THIS ISSUE IN A NOVEL PAPER, CURRENTLY UNDER REVIEW IN JCM. IN THIS CONSENSUS PAPER, SEVERAL OF THE MOST RELEVANT EUROPEAN AND USA NUCLEAR MEDICINE, MRI AND OTHER IMAGING TECHNIQUES EXPERTS DID MEET IN ROME, ADDRESSING KEY QUESTIONS RELATED TO THE ROLE THAT IMAGING SHOULD HAVE IN PJI DEFINITIONS. IF ACCEPTED, THAT PAPER WILL SHED SOME MORE LIGHT ON THE ROLE OF IMAGING TO DIAGNOSE AND DEFINE PJI.

Reviewer 2 Report

This is an interesting study evaluating and validating a novel definition for PJI.

It is the logical consequence of proposing these criteria in 2019.

The only issue I have is that i believe that the manuscript would benefit from a separate paragraph on limitations as there are some.

Inherently, there will be heterogeneity due to the multicentric nature, there is a sparse data bias considering that some tests were only performed in very low numbers and it took six centers and two years to include roughly 200 patients who were evaluated for PJI using these criteria. Compared to other high-volume centers, this is not a whole lot of patients that were included.

This issue could also be adressed in the methods section with respect to the issue how many patients were assessed for PJI in each center using these criteria and how many were treated in general.

Author Response

This is an interesting study evaluating and validating a novel definition for PJI.

It is the logical consequence of proposing these criteria in 2019.

THANK YOU.

The only issue I have is that i believe that the manuscript would benefit from a separate paragraph on limitations as there are some.

STUDY LIMITATIONS ARE NOW REPORTED AS A SEPARATE PARAGRAPH (CF. LINE 308)

Inherently, there will be heterogeneity due to the multicentric nature, there is a sparse data bias considering that some tests were only performed in very low numbers and it took six centers and two years to include roughly 200 patients who were evaluated for PJI using these criteria. Compared to other high-volume centers, this is not a whole lot of patients that were included.

This issue could also be adressed in the methods section with respect to the issue how many patients were assessed for PJI in each center using these criteria and how many were treated in general.

THANK YOU FOR TAKING THIS POINT, THAT GIVES US THE OPPORTUNITY TO CLARIFY THAT, SINCE THE BEGINNING OF THIS RETROSPECTIVE STUDY, THE INPUT HAS BEEN FOR EACH CENTER TO COLLECT AND ANALYZE A SAMPLE OF 30 TO 40 PATIENTS OPERATED BETWEEN JANUARY 2016 AND DECEMBER 2017, THAT MATCHED THE INCLUSION CRITERIA. NO ATTEMPT HAD EVER BEEN MADE TO INCLUDE ALL THE PATIENTS OPERATED IN THAT TIMEFRAME. THIS HAS NOW BEEN MADE CLEAR TO THE READERS BOTH IN THE METHODS (LINE 100) AND IN THE DISCUSSION (LINE 331-335) SECTIONS.